# Investigating differences in the implementation and experience of the Enhanced Health in Care Homes Framework in England: a qualitative protocol for the Understanding Variation in Admissions from Care Homes (UVAC) study

Carl Marincowitz [1], Katherine Elizabeth Zwerger [2], Fawn Harrad-Hyde,[3] Hilary Garrett,[4] Emily Lam,[4] Jennifer Burton [5], Joanne Reeve,[6] Suzanne M Mason [7], Karen Spilsbury [8], David B Price [9], Richard M Jacques [7], Graham Martin [10]

For numbered affiliations see end of article.

**Correspondence to**
Dr Carl Marincowitz;
c.marincowitz@sheffield.ac.uk

## ABSTRACT

**Introduction** Older people living in care homes are at increased risk of harm during acute hospital admissions. In England, care home residents have more than twice as many emergency department (ED) attendances as people of the same age living at home. Up to 40% of emergency hospital admissions of older care home residents may be avoidable with different models of care within their homes. In 2023, National Health Service England introduced the updated Enhanced Health in Care Homes (EHCH) framework, a set of recommendations to support 'joined up' and enhanced care for people living in care homes. A stated aim of the framework is to reduce ED attendances and inpatient admissions of residents. There is limited available evidence regarding how implementation of the EHCH framework differs regionally and whether variation in implementation may impact on hospitalisations of care home residents.

**Aims** We aim to explore regional differences in care elements developed from the EHCH framework and assess how these differences may contribute to variation in hospitalisations of care home residents over the age of 65.

**Methods and analysis** This is a comparative qualitative case study of six care home-containing postcode districts in England embedded within three Integrated Care Boards (ICBs). ICBs are regional organisations responsible for commissioning healthcare services in England. Case study districts and ICBs were selected due to contrasting case-mix adjusted admission rates and other characteristics (eg, deprivation). Data will be collected through semistructured interviews. We will interview health and social care professionals who are responsible for commissioning, overseeing and delivering enhanced care in care homes, care home professionals, residents over the age of 65 and their family and friends. Interview data will be analysed through a framework approach, with comparisons drawn within cases, across cases and across ICBs. Through our analysis, we will characterise the EHCH framework care elements and identify differences in implementation that may cause variation in hospital admissions. We will also identify perceived appropriate, effective and replicable enhanced care models.

Patients and the public have informed the design of this study, and will advise the research practice, support the analysis of data and guide dissemination plans.

**Ethics and dissemination** This study has received Social Care Research Ethics and Health Research Authority Approval (25/IEC08/0014). All participants will be required

## STRENGTHS AND LIMITATIONS OF THIS STUDY

⇒ Our study purposively samples regional case studies using analysis of routine data and characteristics that may impact rates of hospitalisations.
⇒ The three layers of comparison in the case study design will offer in-depth insights in identifying differences in implementation that may cause variation in hospital admissions in different regions.
⇒ We have included a wide range of patient and public involvement and engagement perspectives involved in each stage of the research, including care home residents' perspectives.
⇒ We will not include care home residents who lack capacity to consent to participate in this study, and as a result, the study excludes the perspectives of this resident population who may draw on enhanced care in care homes.
⇒ The study's focus is on enhanced care services implemented for care homes, and not on the quality of care provided within care homes which may also contribute to observed variation in hospital admission rates.

to provide informed consent. The findings will inform a national survey of ICBs to map appropriate and effective enhanced care in England. Findings will be shared with key stakeholders and academic audiences.

## INTRODUCTION

There are over 275 000 people in England and Wales who live in care homes, 82.1% of whom are aged 65 or older.[1] There are multiple risks associated with hospital admissions for frail older people. This can include infections, like hospital-acquired pneumonia, physical deconditioning and delirium.[2 3] In England, there are worsening ambulance response times, waits in the Emergency Department (ED), delays in inpatient admissions and delays in discharging patients from hospital into care.[4 5] Increased length of time in the care journey is associated with increased risk of harm and poorer outcomes.[6] Older people may have a heightened risk of these harms due to pre-existing frailty and the potential deconditioning while in hospital.[7]

In 2015, research by the Health Foundation and Nuffield Trust found that older care home residents had more than twice as many ED attendances as people of the same age who lived at home.[8] More recent research indicates that up to 40% of emergency admissions of older care home residents could be avoidable with different models of care in care homes.[9] The reduction of 'avoidable admissions' is an National Health Service (NHS) long-term priority.[10]

In the *Five Year Forward View* report, published in 2015, NHS England introduced the New Models of Care and 'Vanguards' programme.[11] Among other things, the Vanguards focused on integrating health and social care in care homes through additional provision of specialist and primary care. The programme was implemented with substantial investment in pilot sites, with a focus on co-production of services with local stakeholders.[12] Building from the results of the programme, in 2016, NHS England adopted the Enhanced Health in Care Homes (EHCH) framework, with an update published in 2023.[13 14] The EHCH Framework is a broad set of recommendations with seven core care elements (figure 1).

One of the system-level aims of the framework is to reduce unplanned admissions of care home residents to hospital.[11] In 2023, our update review assessed the effectiveness of interventions adopted internationally to reduce hospital admissions from care homes.[15] UK studies eligible for inclusion in our review predominantly evaluated interventions piloted in the Vanguard programme. Of these eight studies, seven assessed interventions or new care models that increased integration between

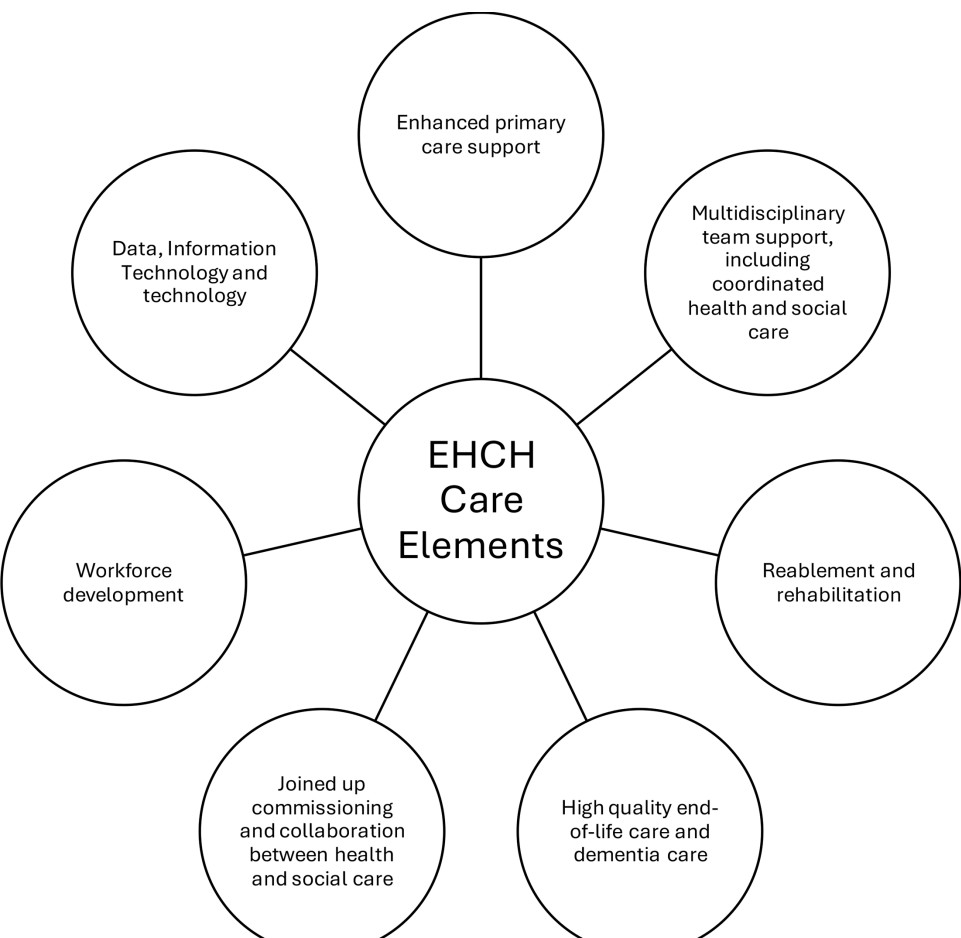

**Figure 1** The seven core care elements outlined in the Enhanced Health in Care Homes (EHCH) framework.

primary and secondary care, and one assessed the impact of enhanced training for care home staff in recognising unwell residents. Our review concluded that hospital admissions from care homes may be reduced by increased care provision by specialist services in care homes that increased integration of care within the urgent and emergency system.[15] This review provides encouraging evidence of the effectiveness of interventions introduced by the Vanguards programme, including integrated working and training and workforce development, but there has been no evaluation of the effectiveness of the subsequent, broader EHCH framework which has been adopted nationally in England. There has been recent research that seeks to characterise the EHCH framework. A study in the South West of England explored person-centred care in care homes, an over-arching component of the whole-system approach of the EHCH framework.[16] This study offers important contributions on characterising person-centred care, highlighting the importance of relationships and inclusion of all those connected to living, working and caring in the care homes. In addition, work by Warmoth and Goodman used a cross-sectional survey to identify and characterise models of primary care commissioned in the East of England as a part of the EHCH framework.[17] They found that implementation of enhanced access to primary care as a part of the framework was limited, with two-thirds of GP respondents reporting no changes to practice. They also found that the approaches to primary care for care homes differed across the region. The findings of this work highlight the need to further characterise what enhanced care is available for care homes and to better understand the barriers and facilitators to implementation in different local and regional contexts.

## Aims and objectives

We aim to explore regional differences in care elements developed from the EHCH framework and assess how these differences may contribute to variation in hospitalisations of older care home residents.

The specific objectives include:

► Undertake comparative, in-depth qualitative case studies of six purposively sampled postcode districts containing one or more care homes embedded within three different ICBs.
► Characterise how the EHCH framework care elements are implemented and used regionally to deepen the existing classification of care elements.
► Identify the differences in elements of enhanced care which may cause variation in hospital admissions between postcode districts containing one or more care homes and ICBs through interviews with commissioners, health and social care practitioners delivering enhanced care and care home professionals.
► Characterise enhanced care services which are perceived to provide appropriate and effective care from the perspectives of care home professionals, care

home residents aged 65 and over (hereafter referred to as older residents) and their family and friends.

## METHODS AND ANALYSIS
### Setting

In England, health and social care services are commissioned by regional organisations, called Integrated Care Boards (ICBs). There are 42 ICBs across the country, each responsible for planning, organising and overseeing the performance of NHS services based on population need. In England, the term 'care home' is an umbrella term that encompasses care settings where people live and receive care and support. There are two main types of care homes. Residential care homes provide personal care, while nursing homes provide both personal and nursing care. As this study is interested in characterising and examining regional differences in the implementation of enhanced care, ICBs, NHS services and care homes are key sites for our investigation.

### Conceptual framework

This study draws on Wennberg's framework of 'unwarranted variation in healthcare', where 'unwarranted variation' reflects supply-sensitive and effective care that has variation in its implementation.[18] An a priori hypothesis of the study is that the different ways in which the EHCH framework care elements are implemented regionally cause observed variation in hospital admission rates at a district and ICB level for older care home residents. This hypothesis was developed from our update review that found that increased provision of specialist care for care homes and further integration of care within the urgent and emergency system through the Vanguards programme may reduce hospitalisations from care homes.[15] This study also draws on work by Appleby et al,[19] who developed a framework mapping the complexity of causes of variation in healthcare. Appleby et al assert that research on healthcare variation should move beyond simply identifying variation to examine the causes of and actions to address it.[19] This study will primarily seek to identify potential causes of variation due to differences in the implementation of the EHCH framework, as well as any care elements within the framework that are perceived as appropriate and effective and could be applied in other regions to address variation. This study also draws on the model of escalation of care home residents to emergency care in hospital developed by Harrad-Hyde et al.[20] The model includes five risk domains that impact internal and external decision-making of care home professionals. In this study, we will use this model to examine how available enhanced care impacts risk management in the decision-making of escalation.

In their stakeholder work to define 'avoidable' hospital admissions, McVey et al[21] found that appropriateness was the concept that best captured the complexity involved in the decision to transfer older care home residents to hospital.[21] Their stakeholders, including residents, their family and friends and care home professionals felt that

appropriateness is a flexible term that supports consideration of individuals' preferences alongside the perspectives of health and care professionals, particularly when compared with 'safety' in the context of avoidable admissions.[21] Informed by their work, this study is focused on the perspectives of residents, their families and friends and health and social care professionals on the appropriateness when residents are not transferred to hospital, particularly in cases (postcode districts) with lower-than-expected rates of inpatient admissions. This study also explores perceptions of effective care, with effectiveness defined by the system-level aim of the EHCH framework to reduce unplanned hospitalisations of care home residents.[14]

### Study design

This study is the second work package of the 'Understanding Variation in Admissions from Care Homes' programme of research. This is a comparative qualitative case study of six care home-containing postcode districts embedded within three ICBs in England. We will apply Yin's embedded multiple case study approach to explore differences in the implementation of EHCH framework care elements and how these differences may contribute to variation in rates of ED attendance and inpatient admissions for older care home residents.[22] Previous similar studies have sampled six case studies to allow a balance between in-depth exploration of issues within and pattern-finding across cases.[22 23] The study will run for eighteen months, from September 2025 to February 2027.

### Sampling of cases and sample size

The cases have been identified in the initial, quantitative work package of our programme of research. Detailed information about this analysis can be found on our study website: https://sites.google.com/sheffield.ac.uk/uvac. The statistical model and the full methods and results are currently under review. For the purposes of this protocol, we outline the key features of this work below.

We used anonymised, routinely collected primary care data to measure how often older care home residents are treated in hospital in different parts of England (sample approximately 20% care home residents across 33 ICBs). Using a multilevel negative binomial regression model, we estimated the number of hospital admissions of older care home residents that would be expected in four 6-monthly periods (2023–2025) in included postcode districts and ICBs when adjusted for resident and care home characteristics. We then compared the actual rates of admissions observed in included postcode districts and ICBs to those that would be expected, estimating funnel plots of observed vs expected hospitalisations of older residents, grouped at the ICB and postcode district level.

Our cases were purposely sampled using positive deviance methods.[24] Positive deviants in this study are care home-containing postcode districts (cases) and ICBs with lower-than-expected rates of hospital admissions of older care home residents. Negative deviants in this study are care home-containing postcode districts (cases) and ICBs with higher-than-expected rates of hospital admissions of older care home residents. The deviants are outliers in our estimated funnel plots (at 95% CIs). In using positive deviance methods to sample cases and ICB with different rates of hospitalisations, we will aim to identify characteristics of enhanced services which may contribute to observed differences in hospital admission rates through our case study work. The funnel plots for included ICBs are available in the online supplemental materials for reference.

Using the funnel plots, we selected three ICBs: one positive deviant, one negative deviant, and one average performer. The negative deviant selected ICB was an outlier in all four 6-monthly periods, with 1.9 to 2.2 times more observed hospital admissions than predicted. The positive deviant ICB was an outlier in two of the 6-monthly periods, with 0.57 to 0.84 times fewer observed admissions than predicted. We further purposely sampled districts as positive and negative deviants embedded within the ICBs. This nested sampling is shown in figure 2.

When sampling ICBs and districts we also incorporated other factors including comparative rurality, age profiles, ethnic diversity and socioeconomic deprivation. The ICB with higher-than-expected rates of hospitalisation is ICB A, with cases 1 and 2 embedded within it.

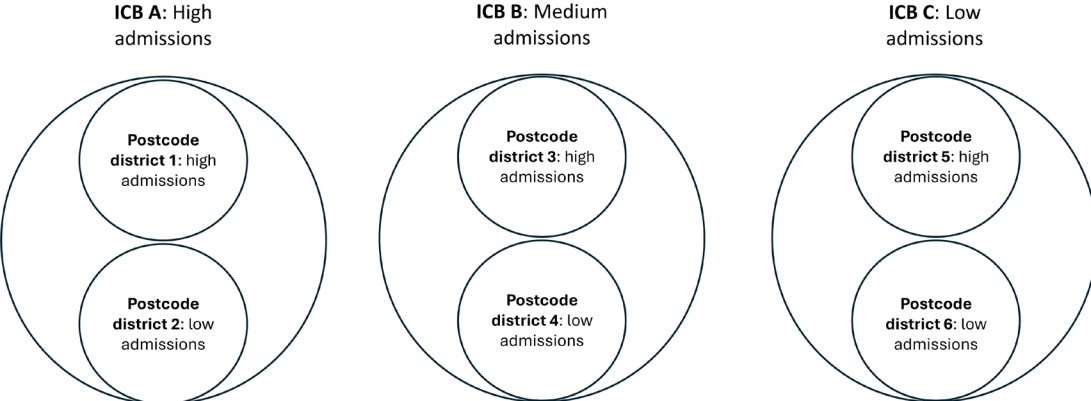

**Figure 2** Sampling of cases embedded within ICBs. ICBs, Integrated Care Boards.

**Table 1** Stages of data collection

| Stage | Participants | Purpose |
|---|---|---|
| 1 | Integrated Care Board (ICB) professionals who have responsibility for overseeing the performance and/or commissioning of enhanced care for care homes, community trust professionals, and GP practices, including GP managers who oversee links with care homes, GPs with responsibilities within care homes and any GP trainee. | ► Identify what enhanced services are commissioned in the area (ICB). <br> ► Identify strengths and weaknesses of these services. |
| 2 | Health and social care practitioners who deliver specialist/enhanced care in care homes, this may include (but is not limited to) clinical practitioners who deliver home rounds, pharmacists who undertake medication reviews, physiotherapists who provide rehabilitation support, dieticians who support nutrition plans and paramedics with extended roles. | ► Characterise commissioned services. <br> ► Identify strengths and weaknesses of services. |
| 3 | Care home professionals, including day and night staff, home managers, nursing staff, shift leaders and carers. | ► Explore experiences of decision-making around escalating care home residents to emergency care. <br> ► Explore the appropriateness and effectiveness of enhanced care. |
| 4 | Older care home residents with an experience of becoming so unwell that they may have needed treatment in hospital in the previous year, and their family and friends. | ► Explore experiences of treatment after becoming very unwell and any enhanced care experienced as a part of this treatment. <br> ► Explore the appropriateness and effectiveness of enhanced care. |

GP, general practice.

ICB A covers a predominately urban area with a population that is younger than the national average and is ethnically diverse. Both case 1 (high-performing) and 2 (low-performing) are in suburban areas. Case 2 has a higher level of household deprivation than case 1. The medium-performing ICB selected for this study (ICB B) covers multiple medium-sized cities and large towns. The proportion of people over 65 in the ICB region is in-line with the national average. Case 3 (high-performing) is in a village, while case 4 (low-performing) is in a large town. Both cases have similar medium-to-high levels of household deprivation. ICB C is largely rural with one medium-sized city and several large towns. The proportion of people 65 or over is higher than the national average and more than 85% of the population is white British. Cases 5 (high-performing) and 6 (low-performing) are both coastal districts with a high number of care homes. Case 5 has a much higher level of household deprivation than case 6. We have therefore selected cases with varying age profiles, levels of rurality, ethnic diversity and socio-economic deprivation to support comparisons within and across cases and ICBs.

### Data collection

Within each of the six cases (sampled districts), we will conduct semistructured interviews with 8–15 participants split across four stages of data collection and analysis (table 1). We will sample participants through purposive and snowball sampling.

### Stage 1

We want to understand what enhanced care is commissioned in the selected postcode districts and ICBs, and the perceived barriers and facilitators to this care. To achieve this, the first stage of data collection will include interviews with health and care professionals who are responsible for commissioning or overseeing the performance of enhanced care in care homes. Based on engagement with our patient and public involvement and engagement (PPIE) and advisory groups, this will include ICB professionals, general practices (GPs) and community/acute trust professionals with managerial roles. We will begin recruiting participants for stage 1 of data collection through our network of stakeholders. The connections made as we conduct each stage will support the recruitment of participants to the subsequent stage.

### Stage 2

In the second stage of data collection, we will further characterise models of enhanced care and the associated strengths and weaknesses by conducting interviews with the health and social care professionals who deliver the elements of enhanced care identified in stage 1. As informed by our PPIE and advisory groups, we anticipate including health and social care practitioners within multi-disciplinary teams who deliver specialist assessments, medication reviews and conduct regular visits to care homes. For this stage individuals must deliver commissioned enhanced care services in care homes that

fall within the EHCH framework care elements. Participants from stage 1 will facilitate identification of potential participants for stage 2.

### Stage 3

In stage 3 of data collection, we will conduct interviews with care home professionals to understand their experiences of escalating residents to hospital and what, if any, enhanced care is a part of their decision-making. We will also explore their perspectives on the appropriateness and effectiveness of EHCH framework care elements. We have been informed by our PPIE and advisory groups to ensure a spread of care home professionals, including day and night shift staff, nurses, carers, managers and shift leaders. For stage 3, individuals must work within a care home and have experience escalating older care home residents to hospital. This will include a variety of professionals, such as care home managers, those who work part-time and full-time hours and those who work at night or during the day.

### Stage 4

We will interview older care home residents with an experience of being unwell enough that they may have needed treatment in hospital in the last twelve months, as well as their family and friends. Through these interviews we will explore their experiences with any EHCH framework care elements involved in their treatment and their perspectives on the appropriateness and effectiveness of that care. Residents will be offered the option of an individual or a dyadic interview with their relative or friend. The stakeholder work by McVey *et al* and engagement with our PPIE groups informed our decision to recruit both family and friends, as this better reflects the reality of support residents may have.[21] Participation in the study will be open to current and former family and friends of older care home residents who had an experience of being unwell enough that they may have needed treatment in hospital in the last twelve months.

To be considered for participation, care home residents must be over the age of 65, live in a care home, have capacity to consent to participate and have had an experience of becoming so unwell that they might have needed to go to hospital in the last twelve months. For family and friends, individuals must be a family member or friend of an older care home resident with an experience of becoming so unwell that they might have needed to go to hospital in the last twelve months to be considered for inclusion in this stage of the study. Individuals who lack capacity to consent to participate will be excluded. The researchers have undergone training on assessing capacity using the principles of the Mental Capacity Act (2005) and will work with care home professionals to assess whether residents have capacity to consent to participate in the study.[25] A standardised tool will not be used to assess capacity. We will also be adopting a process consent approach to informed consent with care home residents, adapted from the work of Dewing.[26 27] This approach will ensure the study team monitor and reaffirm capacity to consent to participate across the research and support care home residents to decide on their participation in a person-centred way.

The stages of data collection will be sequential, with each stage informing the subsequent interviews. The interview guides for stage 1 will be developed using the EHCH framework care elements and focus on participants' professional roles and their understanding of commissioning and delivery of EHCH care elements.

Our conceptual framework will inform our interview guides. We will include questions about awareness of the EHCH framework, what enhanced care is available for care homes and how this care works in practice. The interview guides will also be informed by our a priori hypothesis that, based on the work of Wennberg,[18] the variation in inpatient admissions of care home residents is due to the differences in how enhanced care is implemented. This will guide interview questions on perceptions of effectiveness in appropriately reducing unplanned conveyances of older care home residents to hospital. The interview guides will also be informed by the model of internal and external decision-making in escalating care home residents to emergency care in hospital developed by Harrad-Hyde *et al*.[20] This model includes five risk domains that will frame interview questions on how enhanced care services may impact risk management when escalating a care home resident.

The interview guides will be developed and validated by our PPIE study groups and the study advisory group, whose wide-ranging perspectives reflect those of our participants. The groups will discuss the findings from the previous stages of data collection to develop questions and test the interview guides to ensure they are appropriate for participants in each stage.

We will conduct one semistructured interview with each participant. Participants in stages 1–3 will be offered a Continuing Professional Development certificate on completing the interview. In line with advice from our PPIE groups, we will be providing participating care homes with a voucher or catering an event (such as an afternoon tea) for the residents to thank them for supporting our visits in the home. We will also collect demographic data from participants through equality monitoring forms, based on work by McVey *et al*.[21] Though the population is becoming more diverse, care home residents are predominantly from white backgrounds at 97.5% of the population.[28] In contrast, the adult social care sector workforce is much more diverse, with Skills for Care reporting 63% from white ethnic backgrounds, 11% from Asian/Asian British backgrounds and 18% from Black/African/Caribbean/Black British backgrounds.[29] Our equality monitoring forms will help us ensure we have a representative sample.

Audio recordings of interviews will be transcribed verbatim and anonymised. The study team will share excerpts of anonymised transcripts with collaborators for analysis.

## Data analysis

We will use a framework approach to analyse the interview data, informed by the work of Gale *et al*.[30] Framework analysis is a structured form of thematic analysis that allows for inductive and deductive analysis.

In this study, framework analysis will be undertaken separately for each of the four stages of data collection. This is because the stages of data collection are sequential, with the findings of each informing the development of the interviews for the next stage. The members of the study team (CM, KEZ and FH-H) will lead the analysis. The analysis will begin with line-by-line, deductive coding pertaining to our conceptual framework. EHCH framework care elements will be used to deductively code for typology and participant characterisations of available enhanced care. Our deductive coding will also pertain to our a priori hypothesis and perceptions of appropriate and effective enhanced care. We will draw on the five risk domains from Harrad-Hyde *et al*[20] as deductive codes to examine the potential impact of enhanced care on decision-making for escalating care home residents to hospital. We will also code transcripts inductively, leaving room for any emergent themes and examining participants' views on the commissioning and strengths and weaknesses of available enhanced care in each case.

In undertaking framework analysis, we will analyse a few initial transcripts to develop a set of themes, called a framework, that will then be used when analysing subsequent transcripts. In each stage of data collection/analysis, we will begin with the interviews from both cases embedded within the ICB with medium rates of inpatient admissions of older care home residents. Once we analyse these, the framework will then be applied in analysis of the other four cases (embedded in the ICBs with high and low rates). As each stage builds on and refines the themes of the previous, theoretical saturation, or the point where further data collection does not yield additional insights, will be supported within each case.[31] Framework analysis includes the creation of analytic memos which facilitate cross-checking, and as the research team codes interview transcripts, we will include reflexive noting as a part of these memos. We will be working collaboratively with the relevant PPIE groups in each stage to code parts of transcripts, discuss analytic memos and interpret final themes in workshops. Once the data analysis for all stages is complete, we will also undertake final sense checking with our study advisory group and PPIE groups. We will offer participants an optional, individual meeting to sense check the themes. From our analysis of stages 1 and 2, we will gain a deeper understanding of how the EHCH framework is implemented in each district and what variation exists in terms of available services. Resulting from our analysis of stages 3 and 4, we will understand different perspectives of the appropriateness and effectiveness of models of enhanced care. We will triangulate these perceptions with the admission rates from the analysis in work package 1 to characterise the appropriateness and effectiveness of care associated with the variation observed in our analysis of routine data.

## Reflexivity statement

The members of the study team leading data analysis (CM, KEZ and FH-H) acknowledge that our personal and professional backgrounds and experiences may influence our research practice, including how we analyse the qualitative data. This is especially true regarding our collective biomedical, social care and academic backgrounds. To navigate this, our approach to framework analysis includes layers of cross-checking and collaboration, and we have robust involvement and engagement with our public advisors and stakeholders throughout the study. Our range of empirical and methodological perspectives will also support cross-checking within the study team.

## Limitations

One of the key limitations of this study is the exclusion of care home residents who lack capacity to consent to participate, as this is a population that may draw on enhanced care in their care home. Our decision was made because people who lack capacity to consent to participate in the study due to cognitive impairment may not be able to tell us about their experiences of enhanced care and treatment after being unwell. To mitigate the gap created by their exclusion from participation, we will interview the family and friends of care home residents who lack capacity to consent to participate to understand their perspectives of the enhanced care.

We also discussed incorporating observational methods to mitigate this limitation at length with our PPIE and study advisory groups. Given that the focus of the study is on characterising available enhanced care for care homes in each case, we do not yet know what would be observable frequently enough for observation to constitute an appropriate method of data collection. The groups suggested that without further understanding of what specifically would be observed, the resource requirements from the research team and care homes would outweigh the potential benefit. However, if in the course of the study, we identify specific situations in which it would be appropriate to observe elements of enhanced care, we will adjust the study protocol and seek an amendment to our ethical approval.

A further limitation of the study is the focus on the system-level provision of enhanced care in care for care homes. With this system-level focus, we are not considering the quality of care provided by care homes themselves, which may also contribute to observed variation in hospital admission rates. Similarly, we will not be exploring individual, resident-level factors that may impact hospitalisation (i.e. social relationships and a wider care network).

## Patient and public involvement and engagement

We have multiple PPIE study groups for the duration of the research, covering a diverse range of perspectives. Our

PPIE groups reflect our key participant groups, including older care home residents, family and friends of residents, health and social care professionals with experience working in care homes, and people with experience of urgent and emergency care research and health research for frail older people. Our PPIE groups will continue to be actively involved in this study. They have informed the design of this work package, our approach in recruitment and the public-facing recruitment materials. We will also work collaboratively with our PPIE groups across the data analysis process, with training currently underway to support PPIE members for this work.

### Professional engagement and oversight

We have a study advisory group of health and social care professionals, including perspectives from primary care, geriatrics, ambulance services, third sector organisations and the Care Quality Commission. The advisory group has advised us on the overall design of the study, with specific advice on our approach to recruitment and our recruitment materials. The advisory group will continue to inform the research practice and engage in collaborative data analysis. Our steering committee of topic and methodological experts are responsible for study oversight.

### ETHICS

This study received Social Care Research Ethics and Health Research Authority approval (25/IEC08/0014). The study has a data management in place that describes the deidentification and storage of data in this research, in accordance with General Data Protection Regulation and data protection rules.

Informed consent will be obtained from each participant in this study. For participants in stages 1–3 and family and friends in stage 4, we will obtain written consent. We will adopt a process consent approach, informed by the work of Dewing,[32] in obtaining informed consent with older care home residents. With this approach we will gain an understanding of how each individual consents, obtain verbal consent with an adapted consent form and consistently monitor consent throughout the research interview. We will do so by reminding participants they are able to skip questions, pause and stop the interview and by looking for verbal and non-verbal cues that the person no longer wants to participate. Our approach will also include the co-creation of participant information resources with individuals to suit their communication needs and preferences. In this study, we will only interview those who have the capacity to consent to participate in the research.

### Dissemination and impact

In this study, we aim to create recommendations for appropriate and effective enhanced care services, and how the Enhanced Health in Care Homes framework can be further developed. The findings of this study will be refined by key stakeholders, including the PPIE study groups and the study advisory group through workshops. In these workshops, the stakeholders will determine the best dissemination strategy and format for sharing. Our advisory group includes ICB members, and we will present study findings to the ICBs included in our study. An accessible summary of findings will be shared with the participants, participating care homes and NHS organisations, our PPIE study groups and the Enabling Research in Care Homes Network, who are supporting this study. Dissemination for academic audiences will be undertaken through peer-reviewed papers and presentations at academic conferences. The findings will also inform a national survey of ICBs to map appropriate and effective enhanced care in England in the final work package of the research. We will work with the National Institute for Health and Care Research Yorkshire and Humber Applied Research Collaboration knowledge mobilisation team to further develop our knowledge mobilisation plan, creating a range of dissemination materials aimed at specific stakeholder groups (eg, Department for Health and Social Care commissioners).

**Author affiliations**
[1]ScHARR, The University of Sheffield, Sheffield, UK
[2]Health Services Research, The University of Sheffield, Sheffield, UK
[3]University of Leicester, Leicester, UK
[4]Health Data Research UK, London, UK
[5]Institute of Cardiovascular and Medical Sciences, University of Glasgow, Glasgow, UK
[6]Academy of Primary Care, Hull York Medical School, Hull, UK
[7]School of Medicine and Population Health, The University of Sheffield, Sheffield, UK
[8]School of Healthcare, University of Leeds, Leeds, UK
[9]Optimum Patient Care, Cambridge, UK
[10]University of Cambridge, Cambridge, UK

**Acknowledgements** We would like to thank the Sheffield Emergency Care Forum, Health Data Research UK, Nurturing Innovation in Care Home Excellence Family and Friends group, Alexander Court Care Home and Darnall Dementia Group for their input on this study. We would also like to thank our advisory group for their support: JR, JB, RK, BH, MH, SP, KB, JB, SMM, LC, KR and EL.

**Contributors** CM conceived the study with mentorship from SMM, GM, KS and RMJ. CM, KEZ, EL, HG, FH-H, RMJ, SMM, KS, DBP, JB, JR and GM contributed to the study design and this protocol. All authors contributed to writing the manuscript. CM is the guarantor.

**Funding** CM funded by a National Institute for Health Research (NIHR) Advanced Fellowship (NIHR303605). KEZ and PPIE activities are part-funded by this Advanced Fellowship. This study/research is funded by the National Institute for Health and Care Research (NIHR) Applied Research Collaborations Yorkshire and Humber (NIHR209634). CM NIHR Advanced Fellow, NIHR303605 is funded by the NIHR for this research project. This report is independent research funded by the National Institute for Health and Care Research, Yorkshire and Humber Applied Research Collaborations NIHR200166. GM is based in The Healthcare Improvement Studies Institute (THIS Institute), University of Cambridge. THIS Institute is supported by the Health Foundation, an independent charity committed to bringing about better health and healthcare for people in the UK. This Studystudy is based in part on Datadata from the Optimum Patient Care Research Database (www.opcrd.co.uk) as made available through the Optimum Patient Care Trusted Research Environment under a limited licence granted from Optimum Patient Care and its execution is approved by recognised experts affiliated to the Respiratory Effectiveness Group. However, the interpretation and conclusions contained in this report are those of the authors alone. FH-H is supported by a Mildred Blaxter postdoctoral fellowship from the Foundation for the Sociology of Health and Illness and funding from LOROS Centre for Excellence.

**Disclaimer** The views expressed in this publication are those of the author(s) and not necessarily those of the National Health Service, the NIHR, Department of Health and Social Care or the University of Sheffield.

**Competing interests** None declared.

**Patient and public involvement** Patients and/or the public were involved in the design, or conduct, or reporting, or dissemination plans of this research. Refer to the Methods section for further details.

**Patient consent for publication** Not applicable.

**Provenance and peer review** Not commissioned; externally peer reviewed.

**ORCID iDs**
Carl Marincowitz https://orcid.org/0000-0003-3043-7564
Katherine Elizabeth Zwerger https://orcid.org/0000-0002-1415-7194
Jennifer Burton https://orcid.org/0000-0002-4752-6988
Suzanne M Mason https://orcid.org/0000-0002-1701-0577
Karen Spilsbury https://orcid.org/0000-0002-6908-0032
David B Price https://orcid.org/0000-0002-9728-9992
Richard M Jacques https://orcid.org/0000-0001-6710-5403
Graham Martin https://orcid.org/0000-0003-1979-7577

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
