## [Reviewer comments · BMJ Open]

ARTICLE DETAILS

Title (Provisional)

Investigating differences in the implementation and experience of the Enhanced Health in Care Homes Framework in England: a qualitative protocol for the Understanding Variation in Admissions from Care Homes (UVAC) study.

Authors

Marincowitz, Carl; Zwerger, Katherine Elizabeth; Harrad-Hyde, Fawn; Garrett, Hilary; Lam, Emily; Burton, Jennifer; Reeve, Joanne; Mason, Suzanne M; Spilsbury, Karen; Price, David B.; Jacques, Richard M; Martin, Graham

VERSION 1 - REVIEW

Reviewer	1
Name	Warmoth, Krystal
Affiliation	University of Hertfordshire
Date	24-Nov-2025
COI	None

This protocol outlines a comparative qualitative case study exploring regional differences in implementing the Enhanced Health in Care Homes (EHCH) framework and how these differences may contribute to variation in hospital admissions among older care home residents. The study is well-conceived, addresses a critical issue in geriatric care, and demonstrates strong methodological rigor. The integration of patient and public involvement (PPIE) throughout the design and planned analysis is commendable. The sampling strategy using positive deviance and the embedded multiple case study approach are appropriate for the research aims.

Major Comments

1. The aims and objectives are clearly stated; however, the link between the conceptual framework (Wennburg's unwarranted variation and Appleby's complexity framework) and the practical research questions could be more explicitly articulated. Please add a short paragraph summarising how these frameworks inform the interview topic guides and analysis.

2. The introduction could better situate the study within research on models of care for care homes. Warmoth & Goodman (2022) identified multiple approaches—such as designated GPs for each care home, locality-led care home-specific teams, and multidisciplinary models—while noting persistent variability and slow adoption of EHCH despite national rollout. These findings underscore the importance of understanding implementation challenges and inequities. You may want to reference this evidence to strengthen the rationale for exploring regional differences and clarify how this study builds on or addresses gaps in previous work.

3. The use of positive deviance and funnel plots for case selection is innovative and well-justified. However, the description of how socio-economic and ethnic diversity were incorporated into sampling could be expanded for transparency. Provide more detail on how these factors influenced final case selection beyond “incorporated.”

4. The exclusion of residents lacking capacity is acknowledged as a limitation, but this is a significant gap given the high prevalence of cognitive impairment in care homes. I suggest you consider adding a plan for proxy perspectives or observational data to partially address this limitation. If this is not feasible, please provide a rationale.

5. For the analysis plan, framework analysis is appropriate; however, the description of how inductive and deductive coding will be balanced could be clearer. I suggest you specify whether initial frameworks will be theory-driven (based on EHCH elements or other frameworks) or emergent from data.

6. As you are interviewing those who have the capacity to consent, please clarify how capacity will be assessed and whether any standardised tool will be used.

7. I would suggest you include a reflexivity statement (i.e. how reflexivity will be captured or bias minimised) in the protocol. It is an important component of qualitative research to demonstrate awareness of how researchers’ backgrounds and assumptions may influence data collection and interpretation.

8. The dissemination plan is strong but could include more detail on how findings will influence policy or practice beyond informing a national survey. I suggest you consider a statement on engagement with NHS England or care home networks for implementation.

Minor Comments

- Ensure consistent use of terms such as “enhanced care elements” vs. “EHCH framework elements.”
- Minor typographical errors noted (e.g., “Distrcits” instead of “Districts” in Figure 2 caption).

Comments	Response
1. The aims and objectives are clearly stated; however, the link between the conceptual framework (Wennberg’s unwarranted variation and Appleby’s complexity framework) and the practical research questions could be more explicitly articulated. Please add a short paragraph summarising how these frameworks inform the interview topic guides and analysis.	Thank you for your comment. We now include a paragraph on how the conceptual framework informs the interview guides in the Data collection section on page 9 and our approach in deductive analysis in the Data Analysis section on page 10.
2. The introduction could better situate the study within research on models of care for care homes. Warmoth & Goodman (2022) identified multiple approaches—such as designated GPs for each care home, locality-led care home-specific teams, and multidisciplinary models—while noting persistent variability and slow adoption of EHCH despite national rollout. These findings underscore the importance of understanding implementation challenges and inequities. You may want to reference this evidence to strengthen the rationale for exploring regional differences and clarify how this study builds on or addresses gaps in previous work.	Thank you for highlighting this relevant research. We have added the final paragraph of the introduction on page 3 to highlight these important findings.
3. The use of positive deviance and funnel plots for case selection is innovative and well-justified. However, the description of how socio-economic and ethnic diversity were incorporated into sampling could be expanded for transparency. Provide more detail on how these factors influenced final case selection beyond “incorporated.”	The final paragraph of the sampling section on page 6 now describes in detail how socio-economic and other factors were incorporated into sampling of ICBs and the embedded postcode districts.
4. The exclusion of residents lacking capacity is acknowledged as a limitation, but this is a significant gap given the high prevalence of cognitive impairment in care homes. I suggest	Thank you for raising this important point. We now include a limitations section on page 11 which outlines the reasons for excluding this group and why have not initially included participant observation

you consider adding a plan for proxy perspectives or observational data to partially address this limitation. If this is not feasible, please provide a rationale.	within our protocol. Including observation within the study was discussed at length with our PPIE and advisory groups. If we identify a specific EHCH care element within our case studies which can be practically observed, we will seek to amend our protocol to include this.
5. For the analysis plan, framework analysis is appropriate; however, the description of how inductive and deductive coding will be balanced could be clearer. I suggest you specify whether initial frameworks will be theory-driven (based on EHCH elements or other frameworks) or emergent from data.	We have now expanded the second paragraph of the analysis plan on page 10 to better explain how inductive and deductive coding will be balanced.
6. As you are interviewing those who have the capacity to consent, please clarify how capacity will be assessed and whether any standardised tool will be used.	Stages 1-3 of data collection are with healthcare professionals, and they therefore should have capacity to provide informed consent. For interviews with care home residents in Stage 4, we have added additional details regarding how capacity will be assessed and the use of process consent. This detail can be found on page 8.
7. I would suggest you include a reflexivity statement (i.e. how reflexivity will be captured or bias minimised) in the protocol. It is an important component of qualitative research to demonstrate awareness of how researchers' backgrounds and assumptions may influence data collection and interpretation.	We now include a reflexivity statement on page 11.
8. The dissemination plan is strong but could include more detail on how findings will influence policy or practice beyond informing a national survey. I suggest you consider a statement on engagement with NHS England or care home networks for implementation.	Thank you for your comment-- We have now further expanded our "DISSEMINATION AND IMPACT" section on pages 12-13.
Minor Comments:  • Ensure consistent use of terms such as "enhanced care 	Thank you, these have been standardised and corrected.

elements” vs. “EHCH framework elements.” • Minor typographical errors noted (e.g., “Distrcits” instead of “Districts” in Figure 2 caption).	
--	--

Thank you for your time in reviewing our submission.